# Targeting Copper in Cancer Imaging and Therapy: A New Theragnostic Agent

**DOI:** 10.3390/jcm12010223

**Published:** 2022-12-28

**Authors:** Gabriela Capriotti, Arnoldo Piccardo, Elena Giovannelli, Alberto Signore

**Affiliations:** 1Department of Medical-Surgical Sciences and of Translational Medicine, Sapienza University of Rome, Nuclear Medicine Unit Sant’Andrea University Hospital, 00189 Rome, Italy; 2S.C. Medicina Nucleare Ente Ospedaliero “Ospedali Galliera”, 16128 Genova, Italy

**Keywords:** copper-64, ^64^CuCl_2_, theragnostic, positron emission tomography

## Abstract

Copper is required for cancer cell proliferation and tumor angiogenesis. Copper-64 radionuclide (^64^Cu), a form of copper chloride (^64^CuCl_2_), is rapidly emerging as a diagnostic PET/CT tracer in oncology. It may also represent an interesting alternative to gallium-68 (^68^Ga) as a radionuclide precursor for labelling radiopharmaceuticals used to investigate neuroendocrine tumors and prostate cancer. This emerging interest is also related to the nuclear properties of ^64^CuCl_2_ that make it an ideal theragnostic nuclide. Indeed, ^64^CuCl_2_ emits β^+^ and β^-^ particles together with high-linear-energy-transfer Auger electrons, suggesting the therapeutic potential of ^64^CuCl_2_ for the radionuclide cancer therapy of copper-avid tumors. Recently, ^64^CuCl_2_ was successfully used to image prostate cancer, bladder cancer, glioblastoma multiforme (GBM), and non-small cell lung carcinoma in humans. Copper cancer uptake was related to the expression of human copper transport 1 (hCTR1) on the cancer cell surface. Biodistribution, toxicology and radiation safety studies showed its radiation and toxicology safety. Based on the findings from the preclinical research studies, ^64^CuCl_2_ PET/CT also holds potential for the diagnostic imaging of human hepatocellular carcinoma (HCC), malignant melanoma, and the detection of the intracranial metastasis of copper-avid tumors based on the low physiological background of radioactive copper uptake in the brain.

## 1. Introduction

Currently, basic imaging modalities, such as single photon emission computed tomography (SPECT), positron emission tomography (PET), magnetic resonance imaging (MRI), computed tomography (CT), optical imaging, ultrasound imaging, etc., are regularly used to evaluate specific targets in clinical settings. Among various molecular imaging modalities, the growth of PET imaging has been particularly important over the past decade. PET imaging provides the possibility to evaluate living systems using positron emitting radioisotopes with high sensitivity and specificity. This imaging modality can improve decision making and select only the right patients for tailored therapeutic regimens [1]. A crucial node in molecular imaging with PET is the development of specific radiopharmaceuticals and the choice of the radionuclide to obtain a PET probe, which correlates with its chemical and physical characteristics and feasibility of production. Therefore, copper radionuclides are being studied for both molecular imaging and therapy and many efforts have been made to evaluate the clinical potential of copper radiopharmaceuticals.

## 2. Nuclear Copper and Properties

Research is focused on five of the 32 copper isotopes—^60^Cu, ^61^Cu, ^62^Cu, ^64^Cu, and ^67^Cu—that have the necessary characteristics to be used in the clinical setting (Table 1) [2]. The most studied and promising of these isotopes appears to be ^64^Cu. This isotope decays with the emission of a low-energy positron (650 keV) with shallow average penetration into tissues, that makes this isotope fit for diagnostic PET images with significantly high resolution.

^64^Cu decays through three different routes, namely, electron capture (EC, 42.5%), β^−^ (38%; Eb^-^ 190keV) and β^+^ (19%; Eb^-^ 278keV) decay [Table 1]. Furthermore, ^64^Cu emits Auger electrons. These features make this isotope a pure theragnostic agent [1,3]. The decay characteristics of ^64^Cu allows for PET images that are comparable in quality to those obtained using ^18^F, while the longer half-life (12.7h) of ^64^Cu and its chemical versatility allow for the preparation of many radiopharmaceuticals using different molecules, peptides, proteins and nanoparticles.

^64^Cu can be produced via multiple routes. The most recent approach for the production of copper-64 is by nuclear reactor, which appears to be the most suitable option for regular use in a clinical setting [4]. Cyclotron production is also commonly used. ^64^Ni particles are irradiated with low-energy neutrons, leading to the production of ^64^Cu in the form of [^64^Cu] CuCl_2_, with high purity [3,5].

Regardless of the method of production, ^64^Cu has been used in preclinical and clinical studies that have demonstrated that, in its simple ionic form, it highly accumulates in multiple tumors [5]. In recent years, several studies regarding the use of novel radiopharmaceuticals labelled with ^64^Cu have been published.

## 3. Biological Role of Copper as Theragnostic Agent

Copper is an essential nutrient in mammals, acting as a cofactor in the normal functioning of many physiological processes. It plays an important role in angiogenesis and can stimulate endothelial cell proliferation in a variety of benign and malignant situations. The cellular distribution of copper is rather complex, and although the exact mechanisms by which copper is internalized by human cells are not yet completely clarified, recent data indicate that copper in its ionic form (Cu^2+^) is rapidly bound to plasma proteins in the blood stream (albumin, ceruloplasmin, transcuprein) and reaches the cell surface, where Cu^2+^ ions are reduced to Cu^+^ by specific enzymes called reductases [3,6,7]. Reduced copper enters in the cell through the human copper transporter 1 (hCTR1) [2] and binds to the tri-peptide glutathione (GSH), acting as primary copper acceptor. GSH plays a safeguarding role by binding excess Cu^+^ ions, to prevent oxidative damage due to redox cycling, thus protecting the cell from copper toxicity [8,9]. After this first phase, copper ions are handed off to copper chaperones and proteins (cytochrome c oxidase copper chaperone (COX17), copper chaperone for SOD1 (CCS), and antioxidant protein (ATOX1)), thus keeping them in a bound Cu^+^ state by preventing redox cycling. These chaperones deliver Cu^+^ ions to cytosolic SOD1, cytochrome c oxidase (COX) in the mitochondria and copper transporting ATPase A/B (ATP7A/B) at the trans-Golgi network (TGN), respectively (Figure 1). Analogous mechanisms are carried out by metallothioneins capable of irreversibly binding Cu^+^ ions. When intracellular copper exceeds a certain level, hCTR1 is internalized and destroyed. During this process, copper transporting ATPase A/B (ATP7A and ATP7B) transfer from the TGN to the plasma membrane to help in the excretion of copper from the cell [3,10]. Copper is mainly stored and redistributed by the liver, and thus the hepatobiliary system appears to be the principal means of elimination of excess copper ions [5]. 

Copper ions play an essential role in several biological processes, and copper is a cofactor of many enzymatic reactions, a structural component of different proteins and a key modulator of cell proliferation and growth. Preclinical and clinical studies have demonstrated that copper is deeply involved in the development, growth and progression of malignant lesions. Experimental data have shown significant differences in copper metabolism between normal and cancerous cells. Some of these studies have revealed a significantly higher expression of hCTR1 in malignant tissues than normal tissues, including prostate cancer, lung cancer, glioblastomas, liver cancer, breast cancer, and melanoma [5,11]. Furthermore, copper in its simple ionic form is mostly located in the cytosolic fraction of normal cells, while in tumor cells, these ions mainly diffuse to the nuclear and perinuclear space [3,5]. Therefore, it was hypothesized that hCTR1 could be used as a target for molecular imaging of a wide variety of cancers [8,9,12]. Several researchers have reported an elevated level of copper uptake in malignant tissues [4], such as prostate cancer, lung cancer, breast and liver cancer, glioblastoma and melanoma, and it was proposed that hCTR1 could be used to visualize a large number of tumors [9]. 

An increased copper concentration in the neoplastic cells could be easily evaluated in vivo using radioactive copper (^64^Cu^2+^) ions as radiopharmaceutical. Indeed, ^64^Cu^2+^ ions can serve as an effective biomarker for the noninvasive assessment of cancer, using PET imaging. Many mouse animal models of a wide variety of cancers, such as prostate, lung, breast, ovarian, colorectal, brain and head and neck cancer, hepatoma, fibrosarcoma and melanoma, have been used to explore the biodistribution and uptake of ^64^Cu as a diagnostic or therapeutic agent [9]. Mouse copper transporter receptor 1 (mCTR1) plays the same role as hCTR1 in the intracellular transport of radioactive copper in animal models [9]. PET imaging and biodistribution studies demonstrated the correlation between tumor uptake of ^64^Cu and the level of mCTR1 assessed by immunohistochemistry techniques [4]. Furthermore, preliminary results confirmed that ^64^Cu is selectively concentrated in malignant tissue and not in inflammatory tissues, suggesting the potential role of this radionuclide as a specific tumor marker and its use as a theragnostic agent.

## 4. Cancer Imaging 

As mentioned before, copper is a transitional metal required for the function of many molecules involved in human processes [13,14] and in signaling transduction pathway regulating cancer cell proliferation and tumor growth [15,16,17], giving rise to the possibility to use it for metabolic PET imaging [18,19]. A high intracellular concentration of copper is allowed by an elevated expression of hCTR1, and reduced tumor ^64^Cu uptake and tumor growth inhibition caused by RNA-mediated hCTR1 knockdown have been shown, suggesting that hCTR1 is a promising novel theragnostic target. Some clinical and preliminary human studies have strongly supported these preclinical observations. 

A pilot study was carried out by Capasso E. et al. to evaluate the possible role of ^64^Cu-PET in the staging of patients with prostate cancer (PCa) [20]. Seven patients with PCa were prospectively enrolled, and three patients underwent adrenal deprivation. The remaining patients underwent no therapy. The authors found prostate cancer lesions in the pelvic area, thanks to the absence of urinary excretion of ^64^CuCl_2_. Uptake was higher in the primary tumors of patients without ADT than in treated patients, while the nodal uptake was variable, with focal concentration in normal size node and no significant uptake in suspected lymphadenopathy. These preliminary results indicate the potential role of ^64^Cu-PET for the diagnosis of PCa.

Subsequently, Piccardo A. et al. prospectively evaluated the biodistribution, dosimetry and lesion kinetics in 50 PCa patients with biochemical relapse after surgery or radiation therapy [21]. This study compared the diagnostic performance of ^64^Cu-PET/CT, ^18^F-choline PET/CT, and multiparametric MRI (mpMRI). The results showed the effectiveness of ^64^Cu-PET/CT in detecting local relapse along with bone and nodes metastases (Figure 2). 

The detection rate (DR) of ^64^Cu-PET/CT was higher than the DR of ^18^F-choline PET/CT and multiparametric MRI, particularly in patients with biochemical relapse and a low PSA level [21]. The success of ^64^Cu-PET/CT can be related to the better biodistribution than ^18^F-choline PET/CT; copper ions are not eliminated via the kidneys and do not concentrate in the urinary tract, and this allows an accurate evaluation of the pelvic region and prostatic bed with early visualization of pelvic lesions. Furthermore, dosimetry studies showed that the dose absorbed by PCa recurrences and metastases is low, not considering the therapeutic effect of Auger electrons [22]. More recently, it has been demonstrated that ^64^Cu-PET/MRI shows a higher overall DR in the evaluation of PCa local recurrence than the DR of ^18^F-Choline PET/MRI, ^64^Cu-PET/CT, ^18^F-Choline PET/CT and mpMRI alone [23].

There are relevant findings on potential uses of ^64^CuCl_2_ PET/CT for the study of PCa, but Cantiello F et al. suggested that ^64^CuCl_2_ PET/CT presents some limitations and is not always able to overcome the current imaging methods in use for PCa [24]. In primary PCa staging, there are no differences between mpMRI and ^64^CuCl_2_ PET/CT in metastatic node detection, while in restaging, a significantly higher DR than for 18F-choline PET/CT in the lesion-based analysis both for local and lymph nodal staging can be observed, but not in the patient-based analysis. Furthermore, ^64^CuCl_2_ is mainly removed by the liver, and this could result in a failure to detect liver metastases.

Mascia M. et al. evaluated the safety and efficacy of ^64^CuCl_2_ as a PET radiopharmaceutical to image other urological malignancies [25]. In this prospective study, a total of 23 patients were enrolled, including patients with renal cancer, bladder and penile cancer. Obviously, PCa lesions showed the highest ^64^Cu uptake (SUVmax 11.5), while it was relatively high for bladder cancer (SUVmax 6.2) when compared to penile cancer (SUVmax 3.9) and renal cancer (SUVmax 5.0), suggesting that ^64^Cu-PET/CT might be useful to evaluate primary and local recurrent bladder cancer lesions in view of a high ^64^Cu target to background ratio. In contrast, a high background of ^64^Cu uptake in the kidneys (SUVmax 10.4) could limit its use for the evaluation of primary renal cancer lesions.

Another pilot study has been conducted by Panichelli P. et al. who evaluated the feasibility of ^64^CuCl_2_ PET/CT to image patients affected by glioblastoma (GBM) [26]. A high tumor uptake of ^64^Cu was observed in all patients affected by GBM, in contrast to the low tumor uptake of ^64^Cu in patients diagnosed with low-grade astrocytoma. Remarkably, in the same study, neoplastic tissue was rapidly and clearly detected with stable retention of radioactivity over time. This study provides further evidence to support using ^64^CuCl_2_ as a radiopharmaceutical for PET imaging (Figure 3).

Similar results were obtained by Fiz F. et al., who evaluated the distribution and dosimetry of ^64^CuCl_2_ dosimetry in pediatric patients affected by diffuse high-grade glioma [27]. ^64^Cu-PET/CT presents favorable dosimetry and helps to identify tumor relapses in patients that show unclear MRI results (contrast enhancement and necrosis).

García-Pérez FO et al. evaluated ^64^Cu uptake by non-small cell lung cancer in a limited cohort of patients [28]. The authors observed a high ^64^Cu uptake in peripheral large primary lung cancer lesion on PET/CT images, similar to FDG uptake. High uptake of ^64^Cu was detected in 36% of primary tumors and 27% of nodal metastases. Furthermore, the patients with high tumor uptake of ^64^Cu presented partial response to chemotherapy, while three patients with very low uptake of ^64^Cu displayed disease progression.

As said before, the ^64^Cu uptake is related to the expression of CTR1, thus selecting patients who may benefit from platinum-based therapy, because this transporter could be involved in cellular uptake or the retention of chemotherapy. So, ^64^Cu-PET/CT could be used to avoid ineffective platinum-based chemotherapy in the patients with non-^64^Cu-avid lung cancer lesions.

## 5. Theragnostic Aspect

^64^CuCl_2_ is currently the most actively investigated radiopharmaceutical for both diagnoses and therapy. It has recently been proposed as a promising agent for PCa theragnostics, based on preclinical studies in cells and animal models [29]. Guerreiro et al., using a panel of PC cell lines in comparison with a non-tumoral prostate cell line, performed cytogenetics and radio-cytotoxicity studies to obtain information about cellular consequences to the exposure of ^64^CuCl_2_. In this work, PCa cells were found to exhibit increased ^64^CuCl_2_ uptake, which could not be attributed to an over-expression of copper CTR1 with respect to non-tumoral cells. DNA damage and genomic instability were also high in PCa cells from patients and in tumoral cell lines, exhibiting deficient DNA-damage repair upon exposure to ^64^CuCl_2_.

Pinto C et al. evaluated the potential therapeutic effect in a mouse model of PCa [30]. ^64^CuCl_2_ has significant detrimental effects on cancer cells, being able to reduce their growth and impair the viability. Likewise, the potential theragnostic role of ^64^CuCl_2_ has been evaluated in GBM by Ferrari et al. [31]. The authors demonstrated that ^64^CuCl_2_ exhibits increased affinity for GBM cells than normal cells, thus supporting its potential role as a novel promising diagnostic PET probe for cerebral tumor imaging. Furthermore, their results on therapy suggest that this radiopharmaceutical has great potential as a therapeutic agent for GBM, as clearly shown by the survival curves in experiments carried out with single- and multiple-dose treatments. Catalogna G et al. evaluated the combined effects of ^64^CuCl_2_ and SI113, inhibitor of a serine/threonine protein kinase, on human GBM cell lines [32]. The authors demonstrate that the co-treatment with SI113 leads to synergistic effects on cell death.

At the end of this overview, one more aspect needs to be considered in relation to ^64^Cu ions as radiopharmaceuticals: is the copper toxic? 

Some calculations show that, when used as a radiopharmaceutical, ^64^CuCl_2_ is not harmful to the patient. The total copper in the human body is about 100 mg and the daily dietary intake is about 1–2 mg. It has been reported that the cytotoxic effects of copper ions appear only at concentration ≥ 7.42 mg/L [33,34]. Considering this concentration as the threshold limit, cytotoxicity effects are not considered when using ^64^Cu for PET imaging, because for a PET study, 185–370 MBq of ^64^Cu is usually administered, equivalent to less than 100 µg of Cu^2+^ ions (being the specific activity of ^64^Cu 3.7 MBq/µg^−1^) [20]. In the case of therapy, a dose of 3700 MBq can be administered, being equivalent to 1 mg of Cu^2+^ ions, which is still below the toxicity threshold.

Righi S. et al. [22] demonstrated that the mean dose absorbed by the prostate cancer lesions would be 0.22 Gy for an administered activity of 3700 MBq, suggesting that the therapeutic effect of ^64^Cu may depend on high-linear-energy-transfer (LET) Auger electron emission rather than on the energy released by the beta radiation. Although ^64^Cu may have limited efficacy for large PCa lesion treatment, due to the low average absorbed radiation dose, it may be effective for the eradication of residual disease or for micro-metastasis due to the lethal effect of the Auger electrons emitted by the internalized ^64^Cu in the cell. Furthermore, ^64^Cu represents a potential tool for the radionuclide therapy of other increased metabolic copper tumors including HCC [35], glioblastoma multiforme [31] and malignant melanoma [36].

## 6. Conclusions

In summary, recent advances in clinical trials provide solid evidence to support the potential role of radioactive copper-64 chloride as a useful radiopharmaceutical for cancer imaging by PET. Dosimetric studies in humans demonstrated the safety of ^64^CuCl_2_. Recently, ^64^CuCl_2_ was successfully used for PET imaging of prostate cancer, bladder cancer, glioblastoma multiforme, and non-small cell lung carcinoma.

The potential theragnostic role of ^64^CuCl_2_, due to the high LET electron emission, has been reported and more clinical data are required to confirm the therapeutic efficacy of this radionuclide.

## Figures and Tables

**Figure 1 jcm-12-00223-f001:**
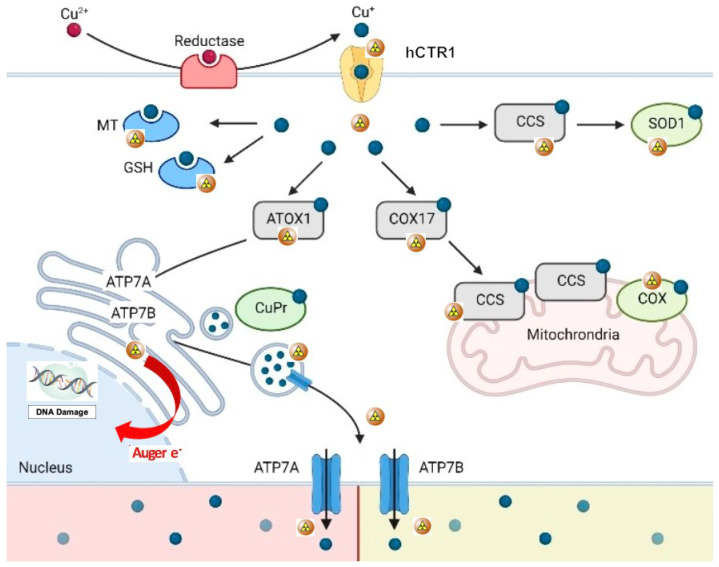
Schematic representation of copper metabolism at the cellular and molecular level (^64^Cu as copper ions). ^64^Cu^2+^ ions are bound to plasma proteins which carry them to the external membrane, where they are reduced to Cu^2+^ by reductases before their uptake into cells. Reduced copper ions are then transported across the cell membrane by the human copper transporter 1 (hCTR1). In the cell, Cu^2+^ions are closely bound by copper chaperones (cytochrome c oxidase copper chaperone (COX17), copper chaperone for SOD1 (CCS), and antioxidant protein (ATOX1)), which deliver copper ions to the cytosol (via SOD1), mitochondria (via COX) and trans-Golgi network (via copper transporting ATPase A/B). Interestingly, glutathione (GSH) binds excess Cu^2+^ to prevent oxidative damage, thus protecting the cell from copper toxicity. Analogous mechanisms are carried out by metallothionein (MT). When intracellular copper is too high, hCTR1 is internalized and destroyed and copper transporting ATPase A/B (ATP7A and ATP7B) transfer from the TGN to the plasma membrane to help in the excretion of copper from the cell (adapted and modified from Michniewicz F. et al.: Copper: An Intracellular Achilles’ Heel Allowing the Targeting of Epigenetics, Kinase Pathways, and Cell Metabolism in Cancer Therapeutics. *Chem Med Chem*
**2021**, *16*, 2315–2329. Copyright Wiley-VCH GmbH. Reproduced with permission).

**Figure 2 jcm-12-00223-f002:**
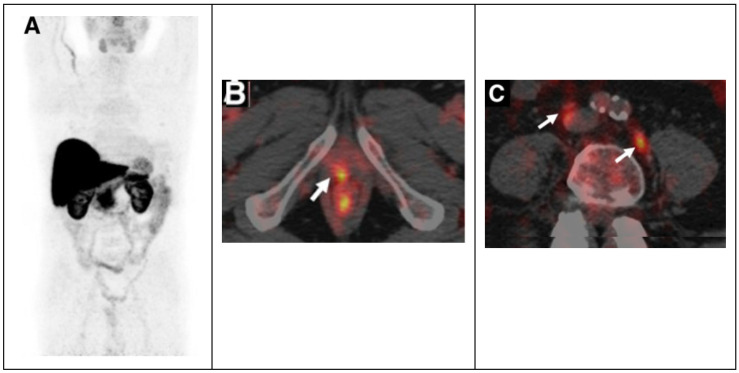
^64^CuCl_2_ PET/CT in prostate cancer relapse. (**A**) ^64^CuCl_2_ biodistribution with high activity in the liver, and kidneys and lesser in the bowel. ^64^CuCl_2_ PET/CT image of pelvis from a old man treated with radical prostatectomy with rising PSA level demonstrated copper-avid recurrence in vescicourethral anastomosis (**B**) and nodes (**C**). Adapted from ref. [21] with (This research was originally published in JNM. A. Piccardo et al. ^64^CuCl_2_ PET/CT in prostate cancer relapse. *J. Nucl. Med.*
**2018**, *59*, 444–451. ©SNMMI).

**Figure 3 jcm-12-00223-f003:**
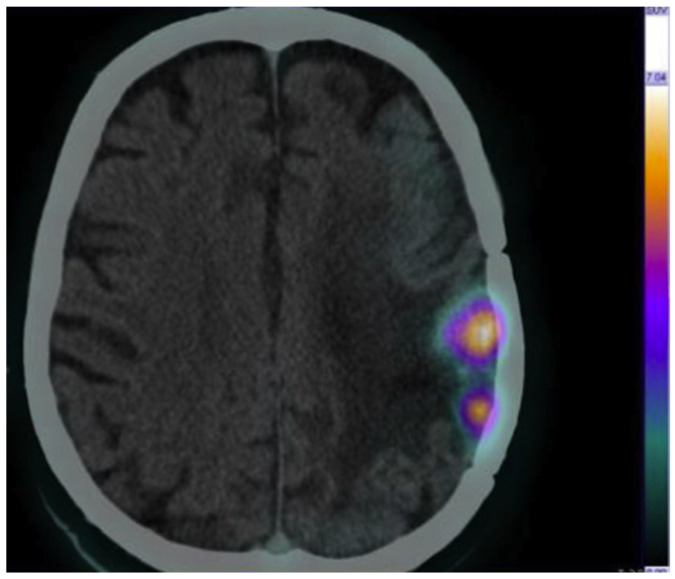
Brain ^64^CuCl_2_ PET/CT scan of a cerebral glioblastoma collected at 1 h post injection (injected activity, 13 MBq/kg). Adapted from ref. [26] with permission of Panichelli P et al. Imaging of brain tumors with copper-64 chloride: Early experience and results. *Cancer Biother. Radiopharm.*
**2016**, *31*, 159–167.

**Table 1 jcm-12-00223-t001:** Decay characteristics and properties of copper radioisotopes.

	^60^Cu	^61^Cu	^62^Cu	^64^Cu	^67^Cu
Production reaction (Source)	^60^Ni(p,n)^60^Cu (Cyclotron)	^61^Ni(p,n)^61^Cu (Cyclotron)	^62^Zn/^62^Cu (Generator) ^62^Ni(p,n)^62^Cu (Cyclotron)	natZn(p,xn)^64^Cu (Reactor) ^63^Cu(n,g)^64^Cu (Reactor) ^64^Ni(p,n)^64^Cu (Cyclotron)	^67^Zn(n,p)^67^Cu (Reactor) ^68^Zn(p,2p)^67^Cu (Cyclotron) ^70^Zn(p,a)^67^Cu (Cyclotron)
Half-life	23.4 min	3.3 h	9.7 min	12.7 h	62.0 h
Decay β^−^ MeV (%)	-	-	-	0.573 (38.4)	0.395 (45) 0.484 (35) 0.577 (20)
Decay β^+^ MeV (%)	3.92 (6) 3.00 (18) 2.00 (69)	1.22 (60)	2.91 (6)	0.655 (17.8)	-
EC	7.4%	40%	2%	43.8%	-
γ MeV (%)	0.511 (186) 0.85 (15) 1.33 (80) 1.76 (52) 2.13 (6)	0.284 (12) 0.38 (3) 0.511 (120)	0.511 (194)	0.511 (35.6) 1.35 (0.6)	0.184 (40)
Clinical Use	Imaging	Imaging	Imaging	Imaging and therapy	Therapy

MeV: megaelettronvolt.

## Data Availability

Not applicable.

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
