# Peer review of "Targeting Copper in Cancer Imaging and Therapy: A New Theragnostic Agent"

_jcm, 2022, doi:10.3390/jcm12010223_

Round 1

Reviewer 1 Report

The authors Dr. G. Capriotti summarized the new theragnostic agent in targeting copper in cancer imaging and therapy. Furthermore, the authors generalized in the field of nuclear copper and properties, biological role of copper as theragnostic agent, cancer imaging therapy, and theragnostic aspect. The authors did a good job for this review.

Reviewer 2 Report

The author has summarized the use of 64CuCl2 in cancer imaging and therapy. They first discussed the advantage of 64Cu as a radioactive isotope and introduced the Cu-involved biological pathways to explain why 64Cu is good for cancer imaging and therapy. Then they summarized examples to show how the 64CuCl2 distributes in patients with prostate cancer, renal cancer, bladder and penile cancer, glioblastoma, and non-small cell lung cancer. At last, they also included a summary of the discussion of how 64Cu can be both diagnostic and therapeutic. 

(1) From line 47 to line 64: the language needs to be reorganized. Most paragraphs here are mainly one or two sentences. 

(2) Line 131: it should be "cancer imaging" as the subtitle. 

(3) From line 158 to line 160: please cite related publications. 

(4) From line 213 to line 223: please try to combine to make them as one or two larger paragraphs. 

Reviewer 3 Report

1. Could you add a paragraph about the possible clinical implications of combining the radionuclide 64CuCl2 with other anti-cancer therapies and the development of a new generation of theranology?

2. Moreover, I would like to ask you about the contraindications for the use of copper radioisotopes in imaging studies?

Reviewer 4 Report

Overall, the information presented by the authors, represents valuable information regarding the role of 64CuCl2 PET/CT tracer in oncology, in both diagnosis and therapy (theragnostics). The half life and its chemical versatility make it suitable for many radiopharmaceuticals preparations. 

Although the article is well written, the abstract needs some improvement, providing more information regarding the role of 64CuCl2.
